# Nonequilibrium phases of ultracold bosons with cavity-induced dynamic gauge fields

Arkadiusz Kosior[1]*, Helmut Ritsch[1], and Farokh Mivehvar[1]

**1** Institute for Theoretical Physics, University of Innsbruck, 6020 Innsbruck, Austria
* arkadiusz.kosior@uibk.ac.at

July 19, 2023

## Abstract

**Gauge fields are a central concept in fundamental theories of physics, and responsible for mediating long-range interactions between elementary particles. Recently, it has been proposed that dynamical gauge fields can be naturally engineered by photons in composite, neutral quantum gas–cavity systems using suitable atom-photon interactions. Here we comprehensively investigate nonequilibrium dynamical phases appearing in a two-leg bosonic lattice model with leg-dependent, dynamical complex tunnelings mediated by cavity-assisted two-photon Raman processes. The system constitutes a minimal dynamical flux-lattice model. We study fixed points of the equations of motion and their stability, the resultant dynamical phase diagram, and the corresponding phase transitions and bifurcations. Notably, the phase diagram features a plethora of nonequilibrium dynamical phases including limit-cycle and chaotic phases. In the end, we relate regular periodic dynamics (i.e., limit-cycle phases) of the system to time crystals.**

# 1 Introduction

Over the past few decades the rapid advancement in experimental techniques for cooling, trapping, and manipulating ultracold quantum gases has allowed one to experimentally realize Feynman's vision of a quantum simulator, i.e, a simple, highly controllable and easy to monitor test system that can be used to mimic key physical phenomena of more complex systems [1–6]. One of the most prominent examples is the realization of artificial gauge fields in ultracold atomic lattices. Gauge fields and potentials play a central role in fundamental physical theories and are responsible for mediating interactions between elementary particles [7]. Although realistic quantum simulation of the central complex problems in particle physics still appears as a far distant goal, much progress has been achieved towards key basic demonstrations of the idea [8,9]. It has been realized that under specific conditions the motion of neutral particles in light fields can mimic the dynamics of charged particles subject to interactions via Abelian and non-Abelian gauge fields [10,11]. Current experimental realizations of synthetic gauge fields are based on ultracold quantum gases in optical lattices manipulated by standard experimental techniques such as the lattice shaking or photon-assisted tunneling [12]. Various experiments have successfully demonstrated the implementation of effective static artificial gauge potentials for ultracold atoms [13–15], mimicking the spin-orbit coupling [16] as well as the density-dependent gauge fields [17]. The range of available opportunities for quantum simulations is presumably even greater and more versatile in driven-dissipative composite quantum-gas–cavity systems [18,19]. As a new branch, many-body cavity quantum electrodynamics (QED) has recently attracted a significant attention from both theoretical and experimental communities [20–24] (cf. Ref. [25] for a review). In contrast to optical lattices where light can be treated as a classical potential, strong atom-photon coupling in optical cavities induces non-negligible atomic back-action on the dynamics of cavity fields. Hence this forms a unique platform to study the physics of dynamical gauge potentials. It has already generated substantial interest in the growing many-body cavity QED community and resulted in the first experimental observation of a dynamic spin-orbit coupling [24] (for theoretical works cf. Ref. [25] and references therein).

Here we continue this path to investigate dynamical density-dependent gauge fields in an ultracold bosonic system, and focus on the reach nonequilibrium dynamical behavior that we encounter. In particular, we focus on a quasi- one dimensional (1D) geometry of a two-component Bose-Einstein condensate (BEC) in a transversely pumped two-mode linear cavity as introduced in our recent Letter [26]. The system can effectively be described by a two-leg Bose-Hubbard model with leg-dependent, dynamical complex tunneling amplitudes generated by cavity-assisted two-photon Raman processes. What is important, the cavity-photon assisted tunneling entails the presence of non-zero atomic currents in our system. With the diagonal tunneling and the atomic pseudo-spin internal states playing the role of a synthetic dimension with two sites, the considered model constitutes a minimal flux-lattice model. As such, it constitutes a first step to the investigation of more complex, dynamical atom-cavity coupled models.

Specifically, in this work we first show that the non-interacting system can be mathematically mapped to a collective spin model, which allows for an efficient numerical and analytical analysis (Sec. 2). Consequently, in Sec. 3 we find and discuss a plethora of nonequilibrium dynamical phases including quasi-stationary, limit-cycle, and chaotic phases. Next, in Sec. 4 we

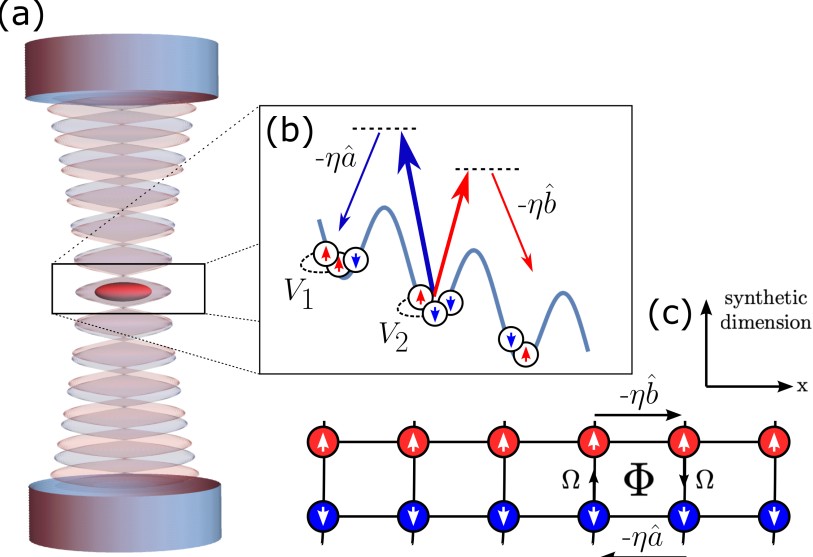

Figure 1: Sketch of the system. (a) A two-component (↑,↓) Bose gas loaded into an external one-dimensional optical lattice [the blue lattice in (b), which is not shown in (a)] perpendicular to the axis of a two-mode optical cavity. (b) Atoms in neighboring sites of the tilted external lattice are Raman coupled with pumping lasers (thick arrows) and cavity fields (thin arrows), applied independently to each atomic components. Consequently, an atomic hopping process is necessarily associated with the emission or absorption of a cavity photon. Atoms interact via two-body contact-interaction potential with the amplitude $V_1$ for the same components and $V_2$ for the opposite components. (c) Schematic picture illustrating geometry and hopping processes of the effective model. The two pseudo-spin components of the BEC can be viewed as synthetic spatial dimensions. Hence, the system is effectively equivalent to a two-leg ladder. The atomic rung tunnelings (i.e., spin-flip processes) are implemented by coupling the two atomic components with a microwave pump with a strength $\Omega$. The directional atomic tunneling along each leg is proportional to the pumping strength $\eta$ and mediated by the cavity fields $\hat{a}$ and $\hat{b}$, that in turn are dynamically coupled to the atomic density [cf., Eqs. (1) and (2), and the discussion below them]. The complex atomic hoppings stemming from the phases of the cavity fields lead a non-zero gauge-invariant magnetic-like flux $\Phi$ piercing each plaquette of the ladder.

support our results with the investigation of the stability of equilibrium points of the derived equations of motion. In Sec. 5 we check the robustness and validity regime of our findings. In Sec. 6 we show that regular, periodic dynamical phases (i.e., limit-cycle states) spontaneously break the continuous time-translation symmetry of the effective Hamiltonian and can be envisaged as time crystals. Finally, in Sec. 7 we present the summary and conclude.

## 2   Physical system and mathematical modelling

Below we first briefly review and summarize the physical model to implement our setup as proposed in our recent Letter [26]. We start from a (pseudo-) spinor Bose-Einstein condensate (BEC) in a transverse 1D optical lattice inside an optical cavity, for which in Sec. 2.1 we derive

an effective Hamiltonian in the form of a two-component Bose-Hubbard model with cavity-photon-assisted complex tunnelings and optomechanical terms. In a second part, in Sec. 2.2, we show that without atomic onsite interactions the Hamiltonian can be mapped to a collective spin problem. In this limit mean-field equations can well approximate the dynamics of the system.

## 2.1 Hamiltonian of the system

Before dealing with the full spinor BEC in a multi-mode cavity model, let us first consider a single-component BEC in an optical lattice, where the atoms can tunnel between neighboring lattice sites [5]. However, due to time reversal symmetry, an atom following a closed trajectory never accumulates a non-trivial geometric (or Berry) phase. Nonetheless it is possible to engineer such non-trivial loop phases in tilted optical lattices, where the suppressed tunneling in the tilt direction is restored via two-photon Raman-laser-assisted hopping with complex amplitude. In other words, both the strength and the phase of the tunneling amplitudes can be experimentally controlled and designed to introduce effective magnetic-like fluxes [11,27,28].

Here, by placing a tilted optical lattice inside an optical cavity with carefully chosen resonances, we can replace one of Raman lasers by a cavity field so that the Raman transition adds or removes a cavity photon per tunneling event. This renders the coupling amplitudes dynamic to obtain atomic state dependent geometric phases [29–31], i.e., a dynamic gauge field generated by atomic currents. When we generalize to a multi-component BEC and a multi-mode cavity, one can then engineer tailored cavity-assisted complex tunnel amplitudes for all atomic components. Mathematically one can treat the pseudospin states as a synthetic dimension [32,33]. As a generic particular example, using this analogy in Ref. [26], we have shown that a quasi-1D, two-component ($\uparrow,\downarrow$) BEC coupled to two modes ($\hat{a}$, $\hat{b}$) of a linear cavity, as shown in Fig. 1, can be viewed as a Bose-Hubbard ladder pierced with a magnetic-like flux $\Phi$. There, we have shown in detail how to implement the atomic rung tunnelings (i.e., spin-flip processes) by coupling the two atomic components with a microwave pump with a strength $\Omega$. The directional atomic tunnelings along the legs are mediated by the cavity fields that are, in turn, dynamically coupled to the atomic density $\bar{n}$. We refer interesting readers to our recent article [26] for the details of the implementation of this model as well as the derivation of the Hamiltonian of the system, and instead here start by only re-expressing the derived Hamiltonian of the system.

Assuming $\hbar = 1$, the effective Hamiltonian of the system reads [26],

$$\hat{H} = \hat{H}_{\text{legs}} + \hat{H}_{\text{rungs}} + \hat{H}_{\text{ph}} + \hat{H}_{\text{a-a}}, \tag{1}$$

with

$$\hat{H}_{\text{legs}} = -\eta \sum_j \left( \hat{a}^\dagger \hat{c}^\dagger_{\downarrow,j+1} \hat{c}_{\downarrow,j} + \hat{b} \hat{c}^\dagger_{\uparrow,j+1} \hat{c}_{\uparrow,j} + \text{H.c.} \right), \tag{2a}$$

$$\hat{H}_{\text{rungs}} = -\Omega \sum_j \left( \hat{c}^\dagger_{\downarrow,j} \hat{c}_{\uparrow,j} + \text{H.c.} \right), \tag{2b}$$

$$\hat{H}_{\text{ph}} = -\left( \Delta - U\hat{N}_\downarrow \right) \hat{a}^\dagger \hat{a} - \left( \Delta - U\hat{N}_\uparrow \right) \hat{b}^\dagger \hat{b}, \tag{2c}$$

$$\hat{H}_{\text{a-a}} = \frac{V_1}{2} \sum_{j,\sigma} \hat{N}_{\sigma,j} \left( \hat{N}_{\sigma,j} - 1 \right) + V_2 \sum_j \hat{N}_{\uparrow,j} \hat{N}_{\downarrow,j}, \tag{2d}$$

where $\hat{c}_{\sigma,j}$ is a bosonic operator annihilating a $\sigma$-spin atom on $j$-th lattice site, $\hat{N}_{\sigma,j} = \hat{c}^\dagger_{\sigma,j} \hat{c}_{\sigma,j}$ and $\hat{N}_\sigma = \sum_j \hat{N}_{\sigma,j}$ are local and total number operators, respectively, $\hat{a}$ and $\hat{b}$ are cavity photon annihilation operators for the two modes. Here, $\eta$ is the pumping strength, $\Delta$ is the pump-cavity detuning, $\Omega$ is the coupling strength between pseudospin components, and $V_1$ and $V_2$

are the strength of intra- and inter-species contact interactions. Note that the effective cavity detunings are dispersively shifted by the optomechanical atomic back-action $U\hat{N}_\sigma$, where $U$ is the dispersive shift per atom. The photon decay rate, $\kappa$, although not included explicitly in Eq. (1), is accounted for in the Heisenberg equations of motions for the cavity-field operators,

$$\partial_t \hat{a} = \frac{i}{\hbar}[\hat{H}, \hat{a}] - \kappa\hat{a}, \tag{3}$$

(and analogously for $\hat{b}$), which is equivalent to the Lindblad-type time evolution of the density matrix in the Schrödinger picture [25].

Although the analysis of Ref. [26] was focused on the stationary regimes of the Hamiltonian (1), we also indicated there the existence of dynamical regimes, such as regular limit-cycle regimes where the amplitudes of the photonic fields follow closed periodic trajectories on a complex plane. Since this dynamical behavior stems from the cavity-induced dynamical geometric phase and the two-body contact interactions do not play any major role [26], in the following we restrict ourselves to the non-interacting regime $V_1 = V_2 = 0$ and set $\hat{H}_{\text{a-a}} = 0$. (For a related recent study on interacting bosons on a two-leg static flux ladder, see Ref. [34].) As we will see in the next section, this assumption allows us to greatly simplify the analysis of the problem. In Sec. 5, we will, however, go back to the fully interacting regime in order to check the robustness of our findings and the validity of the spin model, i.e., we revisit the assumption that small interactions do not mix significantly different quasi-momentum sectors of the Hilbert space.

## 2.2 Spin mapping

In this section we show that the non-interacting bosonic model of Sec. 2.1 can be mapped to a collective spin problem, where a thermodynamic limit can be easily applied. Since the lattice model (1) is translationally invariant, the quasimomentum $k$ is a good quantum number. In the absences of atom-atom interactions, different quasimomenta are not coupled, that is, the quasimomentum is conserved during the time evolution. Therefore, if the system is initially prepared in a single quasimomentum mode, say $k = 0$, the dynamics of the system will be restricted to this single quasimomentum Hilbert subspace.[1] Thus, by taking the Fourier transform of the atomic operator $\hat{c}_{\sigma,j} = \frac{1}{L}\sum_k e^{ikj}\hat{c}_{\sigma,k}$ and restricting to the experimentally relevant $k = 0$ subspace, we readily obtain,

$$\sum_j \hat{c}_{\sigma,j+1}^\dagger \hat{c}_{\sigma,j}^\dagger \Big|_{k=0} = \sum_k e^{ik}\hat{N}_{\sigma,k}\Big|_{k=0} = \hat{N}_{\sigma,k=0}, \tag{4}$$

and similarly for other operators. To further simply the problem, we exploit the standard Schwinger-boson spin representation,

$$\hat{S}_j^x = \frac{1}{2}\left(\hat{c}_{\downarrow,j}^\dagger \hat{c}_{\uparrow,j} + \text{H.c.}\right), \tag{5a}$$

$$\hat{S}_j^y = \frac{i}{2}\left(\hat{c}_{\downarrow,j}^\dagger \hat{c}_{\uparrow,j} - \text{H.c.}\right), \tag{5b}$$

$$\hat{S}_j^z = \frac{1}{2}\left(\hat{N}_{\uparrow,j} - \hat{N}_{\downarrow,j}\right). \tag{5c}$$

After some straightforward algebra the Hamiltonian (1) can be recast as

$$\hat{H} = -2\hat{K}\hat{S}^z - 2\Omega\hat{S}^x + \frac{1}{2}\left(\hat{a}^\dagger \hat{a} + \hat{b}^\dagger \hat{b}\right)\left(U\hat{N} - 2\Delta\right) - \frac{\eta}{2}\left[\hat{N}(\hat{a} + \hat{b}) + \text{H.c.}\right], \tag{6}$$

[1]Although a single mode assumption is experimentally relevant as, usually, before ultracold atoms are loaded into an optical lattice, a BEC is prepared in a spatially uniform ground state in a zero momentum state, we note that by loosening this restriction we expect more interesting physics to appear. For example, a vortex state can exist when the system is prepared in a superposition of two non-equivalent energy minima; see Ref. [26].

with

$$\hat{K} = \frac{U}{2}\left(\hat{a}^\dagger \hat{a} - \hat{b}^\dagger \hat{b}\right) - \frac{\eta}{2}\left[(\hat{a} - \hat{b}) + \text{H.c.}\right]. \tag{7}$$

Here, $\hat{N} = \sum_{j,\sigma} \hat{N}_{j,\sigma} = \hat{N}_{k=0}$ denotes the total particle number operator which, being a conserved quantity, can be replaced by its mean value, i.e., $\hat{N} \to \langle \hat{N} \rangle = N$. Furthermore, the operators $\hat{S}^\gamma = \sum_j \hat{S}^\gamma_j = S^\gamma_{k=0}$, where $\gamma = x, y, z$, represent the components of the collective spin which fulfill the standard spin algebra commutation relations.

Note that if we assume $U = U_0/2L$ and rescale the operators and parameters in the following way,

$$\hat{S}^\alpha \to \hat{S}^\alpha N, \quad \eta \to \eta/\sqrt{N}, \quad \hat{a} \to \hat{a}\sqrt{N}, \quad \hat{b} \to \hat{b}\sqrt{N}, \tag{8}$$

then one can write the new, renormalized Hamiltonian as

$$\hat{H}' = -2\hat{K}'\hat{S}^z - 2\Omega\hat{S}^x + \frac{1}{2}\left(\hat{a}^\dagger \hat{a} + \hat{b}^\dagger \hat{b}\right)(U_0\bar{n} - 2\Delta) - \frac{\eta}{2}\left[(\hat{a} + \hat{b}) + \text{H.c.}\right], \tag{9}$$

with

$$\hat{K}' = \frac{\bar{n}U_0}{2}\left(\hat{a}^\dagger \hat{a} - \hat{b}^\dagger \hat{b}\right) - \frac{\eta}{2}\left[(\hat{a} - \hat{b}) + \text{H.c.}\right], \tag{10}$$

that does not depend on the system size $L$, but on the average atomic density $\bar{n} = N/2L$ only. Consequently, after the rescaling (8), the Heisenberg equations of motion for the spin (i.e., atomic) degrees of freedom take a simple form,

$$\partial_t \hat{S}^x = 2\hat{K}'\hat{S}^y, \tag{11a}$$
$$\partial_t \hat{S}^y = 2\left(\Omega\hat{S}^z - \hat{K}'\hat{S}^x\right), \tag{11b}$$
$$\partial_t \hat{S}^z = -2\Omega\hat{S}^y, \tag{11c}$$

while the equations for the cavity fields read

$$\partial_t \hat{a} = i\left(\hat{\Delta}^a_{\text{eff}} + i\kappa\right)\hat{a} + i\eta\left(\frac{1}{2} - \hat{S}^z\right), \tag{12a}$$

$$\partial_t \hat{b} = i\left(\hat{\Delta}^b_{\text{eff}} + i\kappa\right)\hat{b} + i\eta\left(\frac{1}{2} + \hat{S}^z\right), \tag{12b}$$

with the effective detunings

$$\hat{\Delta}^a_{\text{eff}} = \Delta - \bar{n}U_0\left(\frac{1}{2} - \hat{S}^z\right), \quad \hat{\Delta}^b_{\text{eff}} = \Delta - \bar{n}U_0\left(\frac{1}{2} + \hat{S}^z\right). \tag{13}$$

Since the above equations of motion depend only on the density $\bar{n} = N/2L$, the thermodynamic limit $L \to \infty$, $N \to \infty$ such that $\bar{n} = \text{const.}$ is well defined for the system. Therefore, although the total spin $\hat{\vec{S}}$ is in principle quantized, we can take the thermodynamic limit and effectively treat the spin classically, further simplifying the problem[2]. We also assume that the photonic operators in the thermodynamic limit can be well approximated by their mean-field coherent amplitudes, i.e., $\hat{a} \to \alpha$, $\hat{b} \to \beta$, and consequently $\hat{K}' \to K'$. As long as we are dealing with large photon numbers, an anihilation of a single photon does not change substantially the mean number of photons, and therefore, a coherent state approximation for photons is well justified. It is also true for non- (or weakly) interacting atoms in a thermodynamic limit, where corrections to the mean-field are suppressed as $1/V$ (see, for example, Ref. [25]).

---

[2]Let us note here that the length $|\vec{S}| = [\sum_\gamma (S^\gamma)^2]^{1/2}$ of the total classical spin $\vec{S} = (S^x, S^y, S^z)$ is conserved during the time evolution, being $|\vec{S}| = 1/2$ after the rescaling (8).

Furthermore, let us make an observation that if one defines $\vec{B} = (2\Omega, 0, 2K')$, then the spin equations of motion (11) can be recast in a simple vector form,

$$\partial_t \vec{S} = \vec{S} \times \vec{B}. \tag{14}$$

Consequently, $\vec{B}$ can be considered as an effective magnetic field which, through Eq. (10), explicitly depends on the cavity operators $\hat{a}$ and $\hat{b}$. Although the most interesting scenario is when $\vec{B}$ is time dependent, already in a steady state (i.e., $\partial_t \vec{S} = 0$, $\partial_t \hat{a} = 0$ and $\partial_t \hat{b} = 0$) the relation between spin components $S^z$ and $S^x$ depends in a non-trivial fashion on the stationary values of the cavity modes; see Eq. (11b) and our recent Letter [26].

Let us also stress that the spin mapping allows us not only to significantly decrease the number of degrees of freedom which enormously speeds up the numerical computations, but also it simplifies the analysis of the system [cf. Sec. 3 for the dynamical phase diagram based on the numerical solutions of the equations of motions, and Sec. 4 for the stability analysis of equilibrium points and phase transitions]. In the following sections we will first review stationary phases of the system and then completely focus on the plethora of fully dynamical non-stationary states.

## 3  Dynamical phase diagram

In this section we will analyze the equations of motion, Eqs. (11)–(12), in the thermodynamic limit $L \to \infty$ and $N \to \infty$, keeping $\bar{n} = N/2L = \text{const}$. As we argue in the previous section, in this limit both the components of the total spin and the cavity fields can be treated as classical variables, $\hat{S}^\gamma \to S^\gamma$ and $\hat{a} \to \alpha$, $\hat{b} \to \beta$ respectively, which greatly simplifies the analysis. From now on throughout the article we assume that $\kappa$ determines the energy scale by setting $\kappa = 1$. Also, in order to be consistent with the results of Ref. [26], we take $\Omega = 1$ and $\Delta = U_0 = -6$.

We start our analysis with the numerical study of the long-time behavior of the system, assuming that all atoms are initially in the zero quasimomentum mode and are equally distributed among the two spin components. In the spin language this assumption corresponds to the $\vec{S}(t = 0) = (\frac{1}{2}, 0, 0)$ initial condition for the Heisenberg equations of motion, Eqs. (11). We also assume that initially the photonic fields are only marginally populated by choosing $\alpha$ and $\beta$ as random complex numbers within a circle of amplitude $\epsilon = 0.01$; however, the results do not strongly depend on these initial values.

In order to fully characterize the long-time behavior of the spin model we look at two macroscopic observables: the $z$ component of the total spin

$$\langle S^z \rangle_t = \frac{1}{T_2 - T_1} \int_{T_1}^{T_2} S^z(t)\, \mathrm{d}t, \tag{15}$$

and the vector $\delta \vec{S} = (\delta S^x, \delta S^y, \delta S^z)$, with

$$\delta S^\gamma = \max_{t \in [T_1, T_2]} [S^\gamma(t)] - \min_{t \in [T_1, T_2]} [S^\gamma(t)]. \tag{16}$$

Here, $T_1$ and $T_2$ can be chosen arbitrarily provided that $T_1, T_2, T_2 - T_1 \gg \kappa = 1$. The first observable $\langle S^z \rangle_t$ describes the atomic population imbalance in the two pseudospin components, and is also related to the photon imbalance in the two photonic modes $\Delta n_{\text{ph}} = |\alpha|^2 - |\beta|^2$, cf. Eq. (12). The second observable $\delta \vec{S}$ describes maximal fluctuations in spin components over time. Therefore, the condition $\delta \vec{S} \neq 0$ defines non-steady (non-stationary) states and dynamical phases. The phase diagram of the model, depicted in Fig. 2, involves both stationary and dynamical phases.

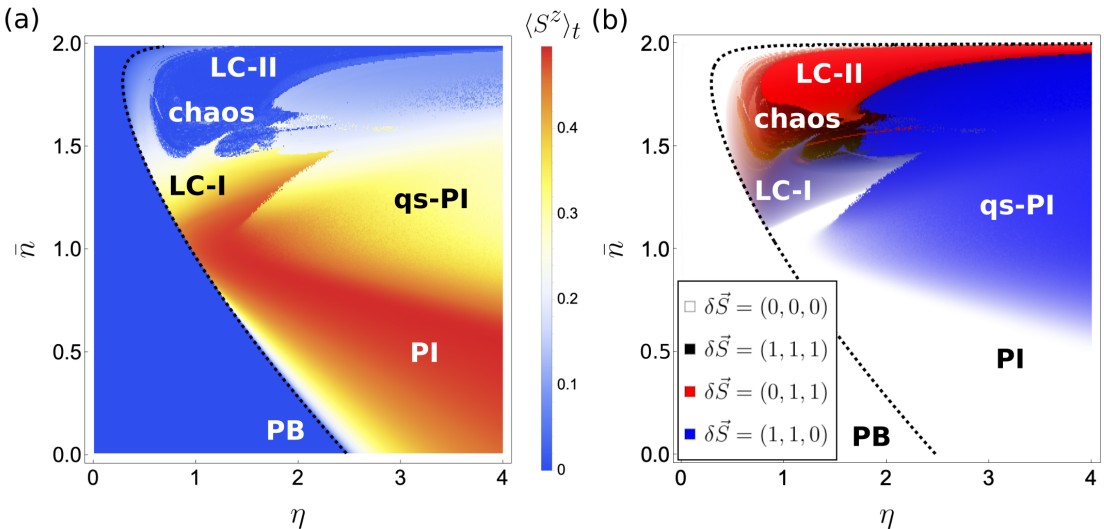

Figure 2: Phase diagram of the system. The long-time behavior of the model can be characterized by two macroscopic observables: the time-averaged, $z$-component of the spin $\langle S^z \rangle_t$ [panel (a)] and the vector of maximum spin fluctuations $\delta \vec{S}$ [panel (b), where CMY color model is used], cf. Eqs. (15) and (16), respectively. Plots reveal two stationary phases—photon balanced (PB) and photon imbalanced (PI) phases—as well as four dynamical phases—limit cycle I (LC-I), limit cycle II (LC-II), quasi-stationary photon imbalance (qs-PI), and chaotic phases. The dynamical phases are characterized in Table. 1; depicted and further explained in Figs. 3–6. The boundary between two stationary phases (black dotted) is calculated analytically according to the Eq. (21). The parameter values are $\kappa = 1$, $\Omega = 1$, and $\Delta = U_0 = -6$ with the initial state $\vec{S} = (\frac{1}{2}, 0, 0)$.

| Phase | Acronym | $\langle S^z \rangle_t$ | $\delta S^x$ | $\delta S^y$ | $\delta S^z$ | Stationary | $\mathbb{Z}_2$ breaking |
|---|---|---|---|---|---|---|---|
| photon balanced | PB | 0 | 0 | 0 | 0 | ✓ | |
| photon imbalanced | PI | $\neq 0$ | 0 | 0 | 0 | ✓ | ✓ |
| quasi-stationary photon imbalanced | qs-PI | $\neq 0$ | $\approx 1$ | $\approx 1$ | $\approx 0$ | | ✓ |
| limit cycle I | LC-I | $\neq 0$ | $\neq 0$ | $\neq 0$ | $\neq 0$ | | ✓ |
| limit cycle II | LC-II | $\approx 0$ | $\approx 0$ | $\approx 1$ | $\approx 1$ | | |
| chaos | | $\approx 0$ | $\approx 1$ | $\approx 1$ | $\approx 1$ | | |

Table 1: Stationary and dynamical phases of the system can be fully characterized by the spin order parameters $\langle S^z \rangle_t$ and $\delta \vec{S} = (\delta S^x, \delta S^y, \delta S^z)$, cf. Eqs. (15) and (16), respectively, as well as Fig. 2. In the phases {PB, PI}, a steady state is reached in a long-time evolution and, hence, $\delta \vec{S}$ vanishes by definition. While in the other phases {qs-PI, LC-I, LC-II, chaos}, $\delta \vec{S}$ is non-zero and the system exhibits a dynamical behavior. All the phases, whether stationary or dynamic, can exhibit spontaneous breaking of the $\mathbb{Z}_2$ symmetry of the model ($S^z \to -S^z$, $\hat{a} \to \hat{b}$), provided that $\langle S^z \rangle_t \neq 0$.

Another important characteristics of the phases of the model is related to the discrete $\mathbb{Z}_2$ symmetry of the spin Hamiltonian (6), being invariant under the simultaneous change

$$S^z \to -S^z, \quad \hat{a} \to \hat{b}. \tag{17}$$

Once $\langle S^z \rangle_t \neq 0$, the $\mathbb{Z}_2$ symmetry of the model is spontaneously broken, which also manifests

itself in the photon imbalance $\Delta n_{\text{ph}} = |\alpha|^2 - |\beta|^2 \neq 0$. The $\mathbb{Z}_2$ symmetry is broken in a stationary photon imbalanced (PI) phase as well as in dynamical, quasi-stationary photon imbalanced (qs-PI) and limit cycle I (LC-I) phases. See Table 1 for the overview and comparison of distinct phases of the system.

## 3.1 Stationary phases

Although in this article our attention is focused on the non-stationary phases of the model, for the sake of self-consistency and completeness let us briefly overview the stationary phases of the model that we characterised in detail (using the atomic description) in Ref. [26]. In a nutshell, there are two distinct classes of steady-state solutions with the respect to the photon-number difference, $\Delta n_{ph} = |\alpha|^2 - |\beta|^2$, belonging to either photon balanced (PB) or photon imbalanced (PI) phases.[3] The cavity fields are uniquely determined by the photonic equations of motions, Eq. (12). In the stationary phases, the cavity field amplitudes, $\alpha$ and $\beta$, obtain non-zero complex phases,

$$\phi_\alpha = -\arctan\left(\frac{\kappa}{\Delta - UN_\downarrow}\right), \quad \phi_\beta = -\arctan\left(\frac{\kappa}{\Delta - UN_\uparrow}\right), \tag{18}$$

that depended nonlinearly on the number of atoms. These time-independent complex phases results in a magnetic-like flux $\Phi = \phi_\alpha + \phi_\beta$ piercing each elementary plaquette of the ladder; see Fig. 1. Notably, even in stationary phases there are non-zero atomic currents along the legs of the ladder flowing in opposite directions, i.e.,

$$J_\downarrow = i\eta \sum_j \langle \hat{a}^\dagger \hat{c}^\dagger_{\downarrow,j+1} \hat{c}_{\downarrow,j} - \text{H.c.} \rangle = 2\kappa|\alpha|^2, \tag{19a}$$

$$J_\uparrow = i\eta \sum_j \langle \hat{b} \hat{c}^\dagger_{\uparrow,j+1} \hat{c}_{\uparrow,j} - \text{H.c.} \rangle = -2\kappa|\beta|^2, \tag{19b}$$

where the dissipation plays an essential role in the generation of these currents; see Ref. [26] for more details.

In the collective spin description that we adopt in this paper (see Sec. 2.2) we distinguish between the PB and PI phases just by looking at the $S^z$ component of the collective spin, which can be related to $\Delta n_{\text{ph}}$ through Eqs. (12). Although the greatest benefits of switching to the spin language will be most evident in the next sections, here we show that already a simple analysis of the stationary equations of motions allows us to find an analytic expression for the PB-PI phase boundary.

From the steady-state equation of motion $\partial_t \vec{S} = 0 = \vec{S} \times \vec{B}$ one trivially obtains $S^y = 0$ and $S^z = -S^x K'/\Omega$. Exploiting the normalization condition for the total spin $|\vec{S}| = 1/2$ leads to

$$K' = -2\Omega S^z / \sqrt{1 - 4(S^z)^2}. \tag{20}$$

Making use of Eq. (10) and expanding both sides of Eq. (20) around $S^z = 0$, we readily obtain the critical value for the strength of the pump $\eta$ across the PB-PI phase boundary,

$$\eta_c = \frac{(\bar{n} - 2)^2 \Delta^2 + 4\kappa^2}{\sqrt{8(\bar{n} - 2)\Delta(\Delta^2 + \kappa^2)/\Omega}}. \tag{21}$$

The analytical curve well matches the numerical boundary between the PB and PI phases; see the dashed black line in Fig. 2. We note that for the chosen initial conditions in Fig. 2,

---

[3]We note that one can further divide the photon balanced (PB) regime into the Meissner (PB-M) and the vortex (PB-V) phases. However, this distinction is not very relevant in the current analysis. We refer interested readers to our recent Letter, Ref. [26].

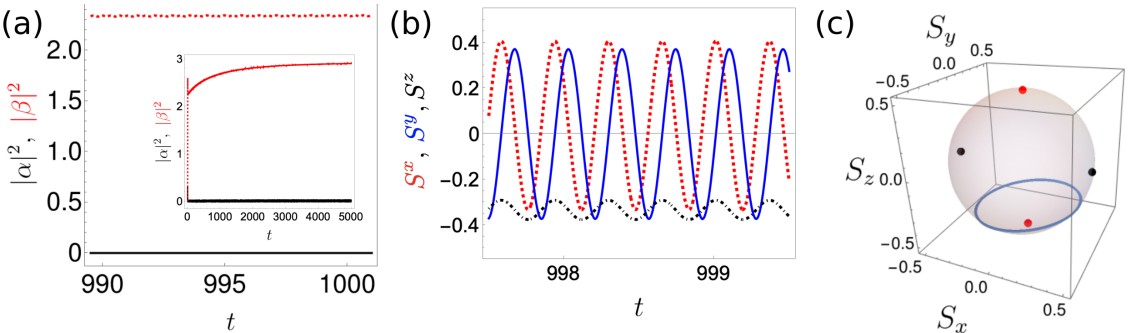

Figure 3: Quasi-stationary photon imbalanced (qs-PI) phase. (a) Long-time evolution of the photonic fields exhibits a photon-imbalanced steady-state-like behavior. However, the qs-PI phase is not stationary as the photons imbalance grows in a long-time evolution as shown in the inset. Although the photon imbalance could in principle saturate reaching the steady state, in the qs-PI phase the saturation time approaches infinity; see the discussion in the main text. (b) The non-stationary character of the qs-PI phase is evident from the behavior of the $S^x$ and $S^y$ components of the collective spin, which perform very fast macroscopic oscillations. (c) The Bloch sphere representation of the total spin which oscillates around one of the $\mathbb{Z}_2$ symmetry-broken unstable equilibrium (red) points of the Heisenberg equations; cf. Sec. 4. (The other two unstable equilibrium points shown in black.) The parameters are fixed at $\bar{n} = 1.3$ and $\eta = 2.2$. The other parameters are the same as in Fig. 2.

the analytical curve of Eq. (21) also reproduces the boundary between the PB and some of the non-stationary phases, which does not need to be true for other initial conditions; see Appendix A.

## 3.2 Non-stationary phases

Although the existence of non-stationary phases was already signaled in our previous Letter [26], in this article we find and quantitatively characterize a plethora of dynamical phases, where the system does not reach a steady state in long-time evolution.

We note that throughout this article we consider the red atom-pump detuning regime, where $U_0 < 0$ and the atoms are high-field seekers which usually leads to steady-state superradiant phases. On the contrary, in the blue detuning regime, $U_0 > 0$ and the particles are repelled from the maxima of the light potential, which can lead to dynamical instabilities such as self-ordered stable limit cycles [35] [see also Refs. [36–39] for the study of limit cycles in a context of time crystals].

Here we find, however, that non-stationary phases can appear for high-field seeking atoms $U_0 < 0$ as long as the effective cavity-pump detunings

$$\Delta_{\text{eff}}^{\sigma} = \Delta \left[ 1 - \bar{n} \left( \frac{1}{2} \pm \hat{S}^z \right) \right], \tag{22}$$

change sign from negative to positive, which may only happen for $\bar{n} > 1$; see Sec. 2.2. In the computed phase diagram shown in Fig. 2, we distinguish four distinct non-stationary dynamical phases: a quasi-stationary photon imbalanced (qs-PI) state, two limit cycle (LC-I, LC-II) phases, and a chaotic phase. All of the dynamical phases, discussed below, are summarized in Table 1 and portrayed in Figs. 3-6.

**Quasi-stationary photon imbalanced phase.** The quasi-stationary photon imbalanced (qs-PI) phase is characterized by fast macroscopic oscillations of the $S^x$ and $S^y$ components of

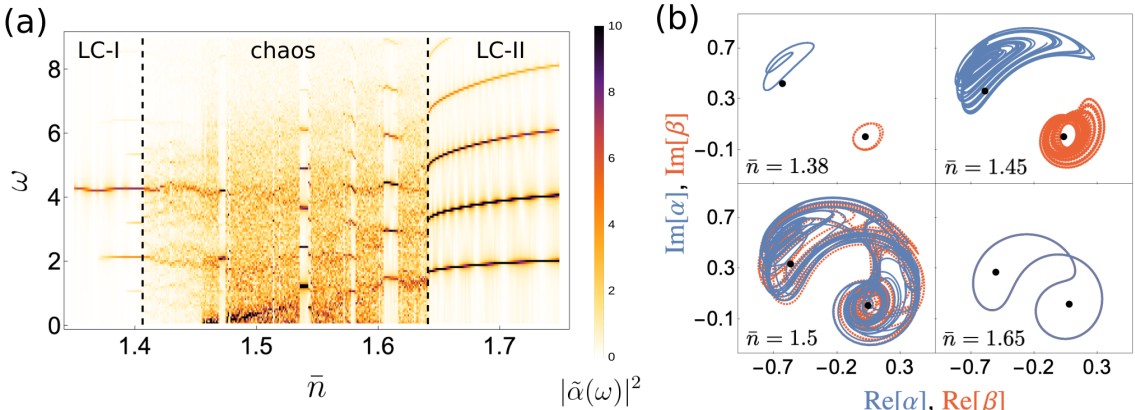

Figure 4: The two limit-cycle phases, LC-I and LC-II, are separated by a chaotic regime. (a) The density $|\tilde{\alpha}(\omega)|^2$ of one of the photon fields in the frequency domain $\omega$ for a fixed value of the pumping strength $\eta = 1.4$. The transition between the LC-I and chaotic phases occurs through the frequency period doubling, where the ratio of consecutive bifurcation intervals is numerically close to the first Feigenbaum constant; see the discussion in the main text. (b) The photonic field amplitudes on the complex plane for different atomic density $\bar{n}$ and the fixed $\eta = 1.4$. The black dots illustrate two repulsive unstable equilibrium points (cf. Sec. 4 for the stability analysis of the equations of motion). As illustrated in panel (a), with increasing $\bar{n}$, one can observe the gradual period doubling of closed limit cycle trajectories in the LC-I phase before entering the chaotic phase. Eventually, after crossing the chaotic regime, the system's dynamics reorganizes and enters the regular LC-II phase. The other parameters are the same as Fig. 2.

the collective spin, i.e., $\delta\vec{S} \approx (1, 1, 0)$; see Fig. 3. By looking solely at the long-time evolution of the photonic fields, it might seem that a (photon-imbalanced) steady state is reached, and therefore, in an experiment the state could be mistaken with the PI phase. Nevertheless, the qs-PI phase is not stationary, since it violates the steady-state spin equation of motion, $\partial_t \vec{S} = 0 = \vec{S} \times \vec{B}$. For this reason the photon imbalance grows in a long-time evolution and/or the photon fields perform small-scale oscillations.

Similar to the PI phase, the qs-PI phase also breaks the $\mathbb{Z}_2$ symmetry of the model. However, as mentioned above, the presence of rapid oscillations of $S^x$ and $S^y$ spin components distinguishes this phase from the stationary PI phase. Furthermore, despite of what we have mentioned before, the qs-PI phase appears for sufficiently large pumping strength $\eta$ already for $\bar{n} < 1$. This simply means that there exists a steady state, $\vec{S} = \pm(0, 0, \pm 1/2)$, but it is not achievable for a finite evolution time [we elaborate on this more in Sec. 4.1]. The phase extends to $\bar{n} > 1$ regime where the amplitude of $S^z$ oscillations increases and the phase continuously changes to a limit cycle phase.

**Limit cycle phases.** The limit cycle phases are characterized by regular oscillations of both spin and photonic fields, cf. Fig. 5. We have identified two limit cycle phases in our model, denoted as the Limit Cycle I (LC-I) and Limit Cycle II (LC-II). In the LC-I phase the total spin performs regular circular oscillations around a mean non-zero value, breaking the $\mathbb{Z}_2$ symmetry of the model. While in the LC-I phase all spin components oscillate, i.e., $\delta S^{\gamma} \neq 0$, in the LC-II phase the variation of $S^x$ is close to zero and $\delta\vec{S} \approx (0, 1, 1)$. Each of the photonic field amplitudes $\{\alpha, \beta\}$ in the LC-I phase follows its own closed regular trajectory on the complex plane. In the LC-II phase, the $\mathbb{Z}_2$ symmetry is not broken as the total spin encircles a big limit-

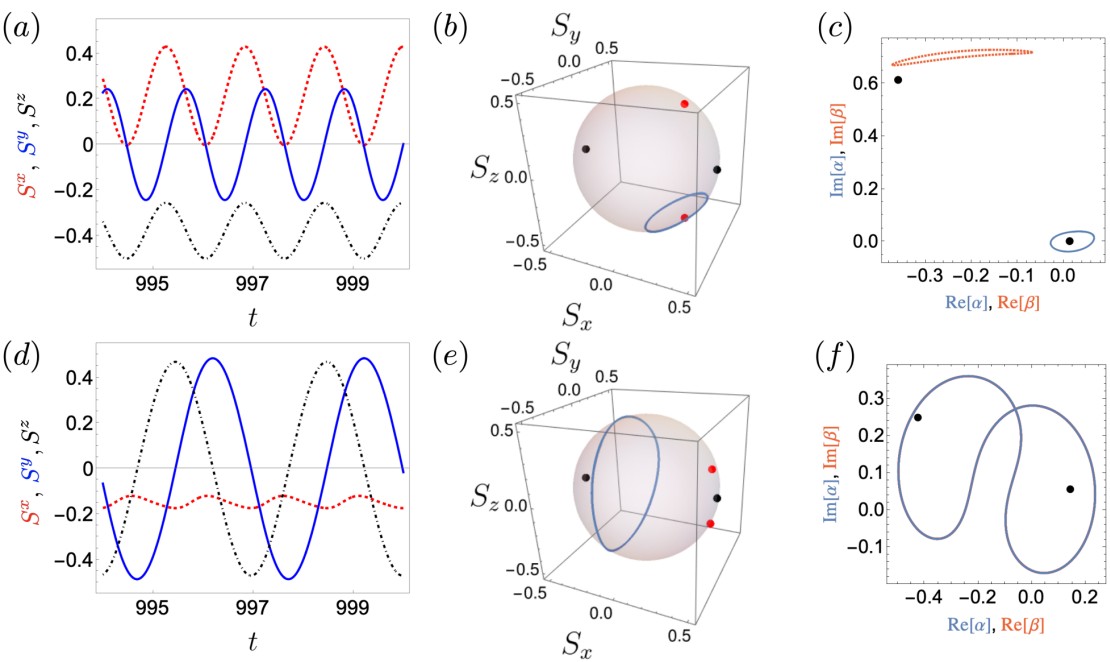

Figure 5: There are two distinct dynamical limit-cycle phases in the system: the limit cycle I (LC-I) phase (top panels, corresponding to $\bar{n} = 1.2$ and $\eta = 0.9$ in the phase diagram of Fig. 2) and the limit cycle II (LC-II) phase (bottom panels, $\bar{n} = 1.85$ and $\eta = 1.4$). (a) The LC-I phase is characterized by macroscopic oscillations of all spin components. (b) The collective spin on the Bloch sphere encircles one of the unstable equilibrium (red) points breaking the $\mathbb{Z}_2$ symmetry of the model. (The other two unstable equilibrium points are represented by black points). (c) Macroscopic oscillation of the $S^z$ component of the total spin leads to periodic oscillations of the photonic field amplitudes $\alpha$ and $\beta$. (d) On the contrary, the LC-II phase is characterized by large-scale oscillations of only two spin components, namely $S^y$ and $S^z$. (e) The LC-II phase does not break the $\mathbb{Z}_2$ symmetry as the collective spin performs big limit cycles, which are stable against the competition between the four repelling unstable equilibrium (red and black) points. (f) Photonic field amplitudes in the LC-II phase evolve along one common closed trajectory. The other parameters are the same as Fig. 2.

cycle trajectory with $\langle S^z \rangle_t \approx 0$. The photonic field amplitudes in the LC-II phase evolve along one common closed trajectory.

**Chaotic phase.** The chaotic phase is characterized by irregular evolution of the spin and the photonic fields [cf. Fig. 6]. The irregular trajectories do not cover the whole Bloch sphere nor the complex plane, but rather form a strange attractor. In Fig. 4 we show the transition between the two limit cycle phases through the chaotic phase. By increasing $\bar{n}$ from inside the LC-I phase and before entering the chaotic phase, one can observe period doubling of closed limit-cycle trajectories. Eventually, the system's dynamics reorganize and enter the regular LC-II phase. The observed period doubling happens only between the LC-I and the chaotic phases. From the numerical simulations we are able to extract the first few period doubling points in the density, $\bar{n}_{\mathrm{PD}} = 1.239, 1.369, 1.392, 1.401$. It is interesting that the numerical value of the weighted average of the ratios of consecutive bifurcation intervals,

$$\langle a \rangle_{\mathrm{av}} = 4.57(46),\tag{23}$$

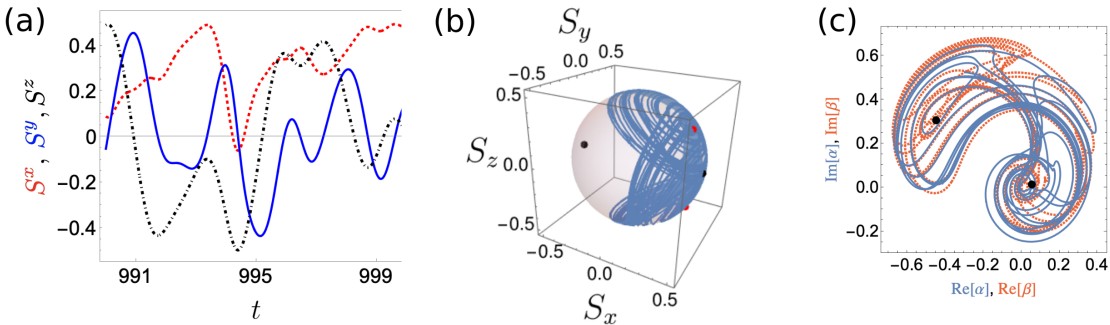

Figure 6: The chaotic phase is characterized by irregular evolution of both the collective spin [panels (a)-(b)] and photonic field amplitudes [panel (c)]. Interestingly, the irregular trajectories do not cover the whole phase-space, rather form strange attractors in a six dimensional space (cf. Sec. 4). The black and red points represent repulsive unstable fixed points. The atomic density and the pump strengths are fixed at $\bar{n} = 1.55$ and $\eta = 1.2$. The other parameters are the same as Fig. 2.

is close to the first Feigenbaum constant, $\delta = 4.669\ldots$, for a bifurcation diagram for a non-linear map [40].

## 4 Stability of equilibrium points

From a mathematical point of view the Heisenberg equations of motion constitute a set of coupled, non-linear autonomous ordinary differential equations. Hence, the stability of stationary solutions can be investigated using standard methods available in nonlinear differential equations textbooks; see for example, Refs. [41, 42]. Here we perform the linear stability analysis by looking at the eigenvalues of the Jacobian matrix, to be discussed in the next subsections. We see that the stability analysis of equilibrium points predicts stationary and non-stationary regimes that agree with the dynamical phase diagram in Fig. 2. In particular, a simple stability analysis of equilibrium points accurately reproduces the phase boundary between the PB-PI phases (Sec. 4.1). Furthermore, it gives us an insight into the Hopf/pitchfork bifurcations and phase transitions which we explain in Sec. 4.2.

### 4.1 Eigenvalues of the Jacobian matrix

In this section we find the equilibrium points of the Heisenberg equations, Eqs. (11)-(12), and investigate their linear stability. Let us start by writing the Heisenberg equations of motion in a compact, vector form as

$$\partial_t \mathbf{X} = \mathbf{F}(\mathbf{X}), \tag{24}$$

with $\mathbf{X} = (S_x, S_y, S_z, \alpha, \bar{\alpha}, \beta, \bar{\beta})^T$ and $\mathbf{F}(\mathbf{X}) = (f_1, f_2, f_3, f_4, f_5, f_6, f_7)^T$, where $f_1 = -2iK'S^y$ is the right-hand side of the first equation of motion (i.e., for $\partial_t S_x$), and similarly for the other components.

The equilibrium points (or steady-state solutions) of Eq. (24) are defined as $\partial_t \mathbf{X}_0 = 0$ and, therefore, can be simply found by solving a set of nonlinear algebraic equations

$$\mathbf{F}(\mathbf{X_0}) = 0. \tag{25}$$

Taking a Taylor expansion (up to the linear term) of the right-hand side of Eq. (24) around

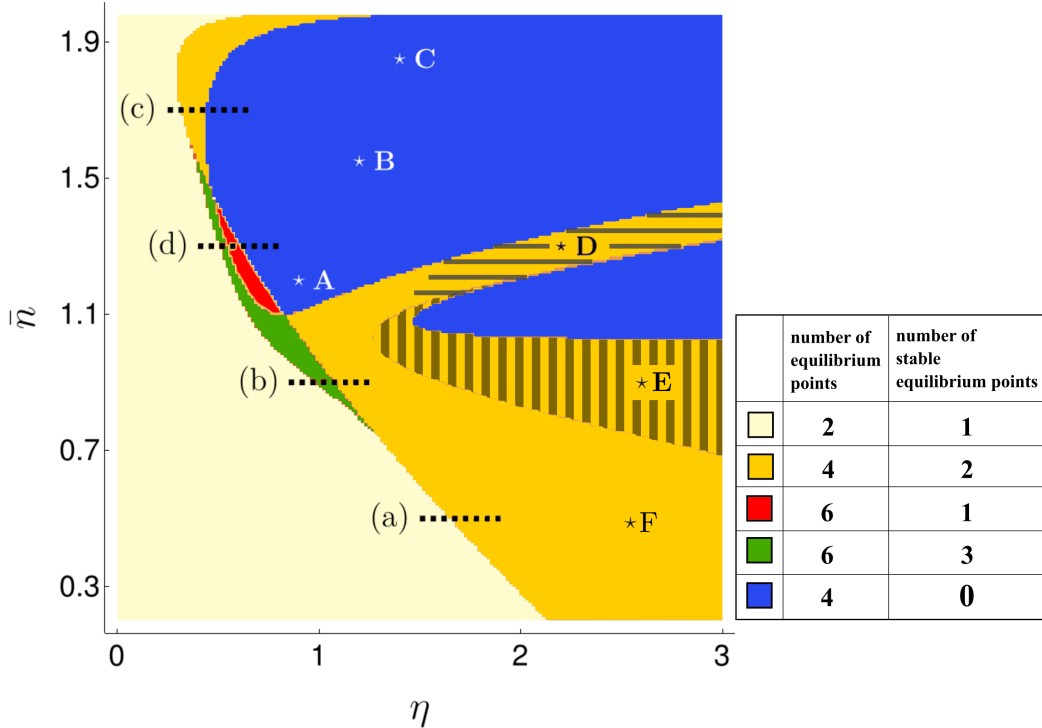

Figure 7: Stability diagram illustrating regions with different number of (stable) equilibrium points. The density plot is in a good agreement with the numerical phase diagram of the system, Fig. 2. The lines (a)-(d) corresponds to panels in Fig. 8, which shows different bifurcation scenarios at phase boundaries. Capital letters correspond to the table of equilibrium points; see Tab. 2. Note that although the vertically striped area has two stable equilibrium points, dominant eigenvalues of each of the stable points has a very small negative real part close to zero, $|\text{Re } \lambda| \leq 10^{-4}$. Consequently, none of the stable equilibrium points cannot be reached for experimentally relevant evolution times; cf. Fig. 2 and Tab. 2. In a similarity, the horizontally striped area has also two stable equilibrium points, but their energy is high and those point are not reached dynamically for chosen initial conditions.

the equilibrium point $X_0$ yields

$$\partial_t \mathbf{X} = \mathbf{F}(\mathbf{X_0}) + \left.\frac{\partial \mathbf{F}(\mathbf{X})}{\partial \mathbf{X}}\right|_{\mathbf{X_0}} (\mathbf{X} - \mathbf{X_0}). \tag{26}$$

Since $\partial_t \mathbf{X_0} = 0$, Eq. (26) can be readily recast in the form,

$$\partial_t \delta \mathbf{X} = \mathbf{J_0} \delta \mathbf{X}, \tag{27}$$

where $\delta \mathbf{X} = \mathbf{X} - \mathbf{X_0}$ and $\mathbf{J_0} = \left.\partial_{\mathbf{X}} \mathbf{F}(\mathbf{X})\right|_{\mathbf{X_0}}$ is the Jacobian matrix. A general solution of Eq. (27) can be written as a superposition of terms of the form $\exp(\lambda_j t)$, where $\{\lambda_j\}$ is the set of eigenvalues of the Jacobian $\mathbf{J_0}$. Hence, the problem of stability of equilibrium points as well as dynamics of the system in the vicinity of the equilibrium points is reduced to the analysis of a few eigenvalues of the Jacobian. In particular, the equilibrium points are stable (i.e., the equilibrium points are attractive) if and only if all the eigenvalues of the Jacobian have negative real parts,

$$\forall_j \text{Re}[\lambda_j] < 0. \tag{28}$$

Moreover, one can gain an important insight into the properties of the set of eigenvalues of the Jacobian $\mathbf{J_0}$ just by looking at the symmetries of $\mathbf{F(X)}$.

In our model, because of the conservation of the total spin $|\vec{S}| = \frac{1}{2}$, there are only six independent equations and, therefore, one eigenvalue of $\mathbf{J_0}$ is identically zero. We can discard $\lambda = 0$ from our analysis as long as we restrict ourselves to a constant spin hypersurface. Furthermore, one can make a simple observation that

$$\mathbf{F(X)} = \left[ \mathbf{U^\dagger F(X) U} \right]^*, \tag{29}$$

with $\mathbf{U}$ being a unitary, block diagonal matrix,

$$\mathbf{U} = \begin{pmatrix} \mathbb{1}_3 & 0 & 0 \\ 0 & \sigma_x & 0 \\ 0 & 0 & \sigma_x \end{pmatrix}, \tag{30}$$

where $\mathbb{1}_3$ and $\sigma_x$ denote the identity matrix of size 3 and the first Pauli matrix, respectively. Using Eq. (29) we can immediately infer that the remaining eigenvalues of $\mathbf{J_0}$ are either purely real or come in complex conjugate pairs $\lambda_i$ and $\lambda_i^*$. In the latter case, if the complex conjugate pair has a positive real part, then the equilibrium point is not stable. However, a stable limit cycle could appear in its vicinity.

In Fig. 7 we plot the stability phase diagram of the model. Comparing the stability diagram with the dynamical phase diagram of Fig. 2 reveals that the two phase diagrams are in good agreement. In particular, the phase boundary between the two stationary phases (PB and PI) is perfectly reproduced by a line separating regions with one and two stable equilibrium points. The blue region in Fig. 7, i.e., the region without stable equilibrium points, corresponds to dynamical phases in Fig. 2. However, in order to recognize different dynamical phases, one must analyze the behaviour of eigenvalues of the Jacobian corresponding to the equilibrium points; see Tab. 2. In particular, the LC-I phase (in a vicinity of the point A in Fig. 7) can be distinguished by looking at the dominant eigenvalues, i.e., eigenvalues with the largest real part, for the two $S_z \neq 0$ equilibrium points. Namely, in the LC-I phase the real parts of the dominant eigenvalues are small in comparison with a distance on the Bloch sphere between the two equilibrium points.

A major difference between Fig. 2 and Fig. 7 is the existence of regions with stable equilibrium points in the dynamical qs-PI phase. This behaviour can be again explained by analysing the dominant eigenvalues and mean energies at the equilibrium points. In the yellow region in the vicinity of the point D in Fig. 7 there exist two stable equilibrium points with non-zero $S^z$, but their energy is much higher than the energy of other unstable equilibrium points; see Tab. 2. For this reason, the stable equilibrium points are not reached in the time evolution unless the initial conditions are fine-tuned to the vicinity of one of the stable points. In the striped region in Fig. 7, on the other hand, there are two stable equilibrium points with minimal energy, but the real part of the dominant eigenvalue pair is so small ($|\mathrm{Re}\lambda| \leq 10^{-4}$) that the stable points cannot be reached in a finite, experimentally relevant evolution time. This indicates that there is a continuous crossover between the PI and qs-PI phases.

Finally, in Fig. 7 there exist also narrow red and green regions with six equilibrium points. In Sec. 4.2, we will see that the existence of such region entails the appearance of subcritical pitchfork and Hopf bifurcation of solutions [41–43], which we discuss below.

## 4.2  Phase transitions and bifurcations analysis

In this section we focus on bifurcation of solutions across the phase boundary between the stationary PB-PI phases as well as between the PB and dynamical limit cycle phases.

| | $S^x$ | $S^z$ | $\langle \hat{H}' \rangle$ | Re $\lambda$ | | $S^x$ | $S^z$ | $\langle \hat{H}' \rangle$ | Re $\lambda$ |
|---|---|---|---|---|---|---|---|---|---|
| **A** | -0.50 | 0.00 | 0.86 | 0.85 | **D** | -0.50 | 0.00 | 0.06 | 2.08 |
| | 0.28 | -0.41 | -0.27 | 0.09 | | 0.16 | -0.47 | 1.74 | -0.10 |
| | 0.28 | 0.41 | -0.27 | 0.09 | | 0.16 | 0.47 | 1.74 | -0.10 |
| | 0.50 | 0.00 | -1.14 | 0.83 | | 0.50 | 0.00 | -1.94 | 3.06 |
| **B** | -0.50 | 0.00 | 0.66 | 1.48 | **E** | -0.50 | 0.00 | 0.06 | 1.83 |
| | 0.40 | -0.30 | -0.38 | 0.82 | | 0.03 | -0.49 | -3.05 | -0.00 |
| | 0.40 | 0.30 | -0.38 | 0.82 | | 0.03 | 0.49 | -3.05 | -0.00 |
| | 0.50 | 0.00 | -1.34 | 2.16 | | 0.50 | 0.00 | -1.94 | 2.70 |
| **C** | -0.50 | 0.00 | 0.63 | 1.58 | **F** | -0.50 | 0.00 | 0.38 | 0.90 |
| | 0.46 | -0.19 | -0.57 | 0.83 | | 0.24 | -0.44 | -1.86 | -0.50 |
| | 0.46 | 0.19 | -0.57 | 0.83 | | 0.24 | 0.44 | -1.86 | -0.50 |
| | 0.50 | 0.00 | -1.37 | 2.35 | | 0.50 | 0.00 | -1.62 | 1.51 |

Table 2: Table of all equilibrium points for the points A-F in Fig. 7, their energy $\langle \hat{H}' \rangle$ and real value of the dominant eigenvalue Re $\lambda$ (i.e., with the maximal real part). Note that the equilibrium points always have $S^y = 0$, so they can be unambiguously labeled by $S^x$ and $S^z$ spin components.

By definition, a bifurcation is a change of the topological type of the system, e.g. the number of equilibrium points, as its parameters pass through a critical bifurcation value (see, for example, Ref. [41–43]). For instance, in the pitchfork bifurcation, which is probably the most commonly appearing bifurcation type in physical systems, two new equilibrium points appear at the critical point while the fixed equilibrium changes its stability. Pitchfork bifurcations (and also Hopf bifurcations which lead to the appearance of periodic oscillations rather then new equilibra) have two types – supercritical and subcritical. In Fig. 8 we illustrate four different types of bifurcation that we observe in our system. Panels (a)–(d) of Fig. 8 correspond to the lines in Fig. 7, respectively. In particular, Fig. 8(a) shows a standard supercritical bifurcation, where at the critical $\eta_c$ the stable $S^z = 0$ equilibrium point looses its stability and two new branches of stable solutions with $S^z \neq 0$ appear. This bifurcation corresponds to a second order phase transition where the order parameter scales as

$$S^z(\delta\eta = \eta - \eta_c) = \begin{cases} 0 & , \quad \delta\eta \leq 0 \\ (\delta\eta)^\nu & , \quad \delta\eta > 0 \end{cases}. \tag{31}$$

The value of the critical exponent is found numerically to be $\nu \approx 0.5$, which is in an agreement with the standard Landau theory of phase transitions [44]; see Fig. 9.

On the contrary, Fig. 8(b) shows an example of a subcritical pitchfork bifurcation which is responsible for the first order phase transition between the PB-PI phases. In a subcritical pitchfork bifurcation, despite that at $\eta_c'$ two new stable branches of solutions with $S^z \neq 0$ emerge, the $S^z = 0$ solution loses its stability only at the critical point $\eta_c > \eta_c'$. Hence, the order parameter exhibits a discontinuous jump, a characteristic of a first order phase transition.

Finally, Fig. 8(c)–(d) depict Hopf bifurcations across the phase boundaries between the PB and limit cycle phases. Although at a first glance, the solution branching at the Hopf bifurcations is very similar to the behavior of the pitchfork bifurcations, there is an important difference that at $\eta_c''$, the $S_z \neq 0$ equilibrium points lose their stability. That means that there is no stable stationary solution. However, new time-dependent solutions emerge. The behavior of eigenvalues of the Jacobian at the Hopf bifurcations are depicted in Fig. 10. Specifically, Fig. 10(a) shows eigenvalues of the Jacobian for one of the $\mathbb{Z}_2$ symmetry broken equilibrium points with non-zero $S^z$. Below $\eta_c''$ all eigenvalues have negative real part and, therefore, the

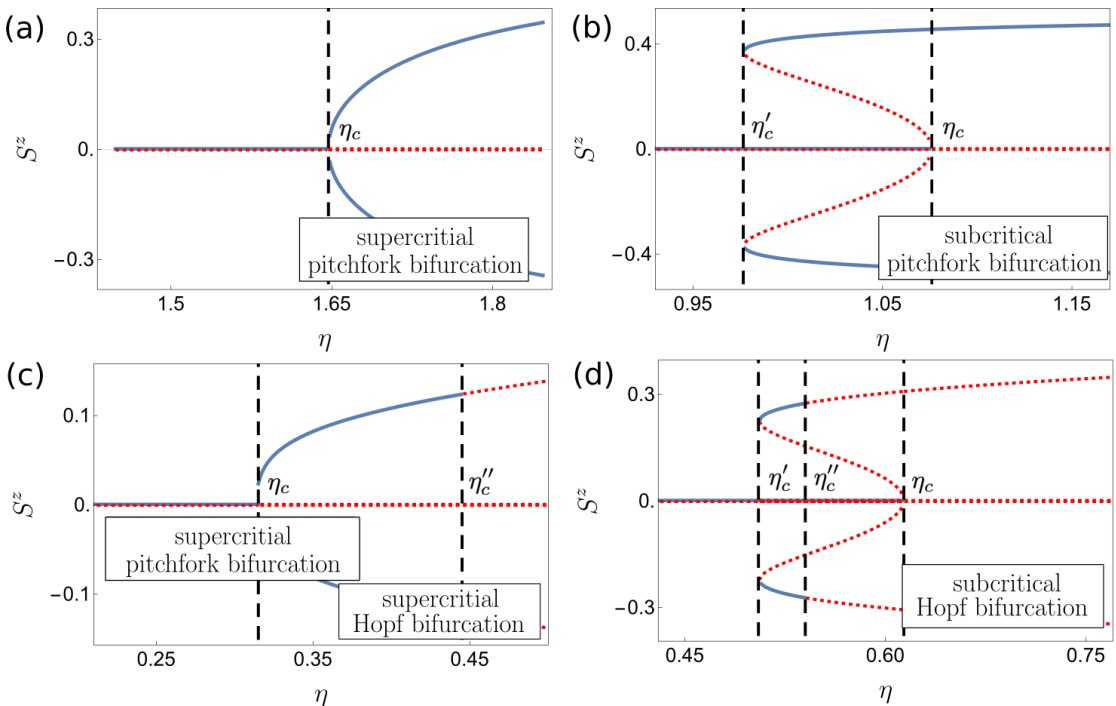

Figure 8: Nature of bifurcations across the phase transitions. Panels illustrate the behavior of the equilibrium points as a function of $\eta$ for a fixed density $\bar{n}$. Panels (a)–(d) correspond to the in Fig. 7. The stable (unstable) solution branches are represented with blue solid (red dotted) lines. Four different bifurcation types are present in the system. (a) A supercritical pitchfork bifurcation ($\bar{n} = 0.5$). At the critical point $\eta_c$, the $S^z = 0$ stable equilibrium point loses its stability, but two new $S^z \neq 0$ stable equilibrium points emerge. (b) A subcritical pitchfork bifurcation ($\bar{n} = 0.9$). Although at $\eta'_c$ two pairs of $S^z \neq 0$ stable and unstable branches of solutions emerge, the lowest energy equilibrium point (corresponding to $S_z = 0$) loses its stability only after crossing the critical point $\eta_c$. Subcritical pitchfork bifurcations are characteristic of first-order phase transitions. (c)–(d): Supercritical and subcritical Hopf bifurcations ($\bar{n} = 1.7$ and $\bar{n} = 1.3$, respectively). The panels are similar to (a)–(b), with an important difference that at $\eta''_c$, the $S_z \neq 0$ equilibrium points lose stability. At the critical point $\eta_c$ in panel (d) the system jumps to one of the stable, dynamical limit cycles.

equilibrium point is stable. At the Hopf bifurcation a pair of complex-conjugate eigenvalues cross the imaginary axis. Thus, the stable equilibrium point becomes unstable, but a periodic orbit, i.e. a limit cycle, appears. Furthermore, Fig. 10(b) illustrates eigenvalues of the Jacobian for an unstable equilibrium point (blue diamonds) corresponding to the $S^z \neq 0$ order parameter, as well as eigenvalues of the Jacobian for the $S^z = 0$ equilibrium point (red disks). Once the latter loses its stability, the solution jumps to one of the stable limit cycle solutions.

## 5 Validity of the spin model

In this section we take a step back and consider again the full bosonic model, Eq. (1), in order to investigate the validity and the robustness of our results. In Sec. 2.2, we assumed that the atoms do not directly interact with each other (non-interacting, ideal bosons) and mapped the bosonic ladder system to the collective spin problem. By doing so, we restricted our analysis

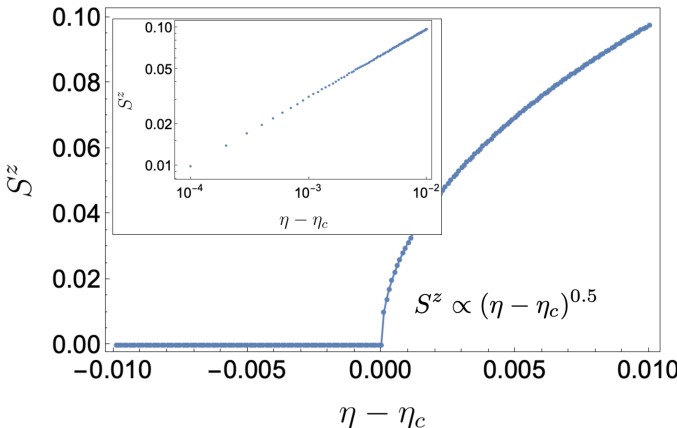

Figure 9: Supercritical pitchfork bifurcation responsible for a second order phase transition between the PB and PI phases. The $S^z$ spin component changes algebraically $S_z \propto (\eta - \eta_c)^\nu$ after crossing the critical point $\eta_c$, which is also illustrated on a log-log plot in the inset. Note that the fitted critical exponent $\nu \approx 0.5$ is the same as in Landau theory [44].

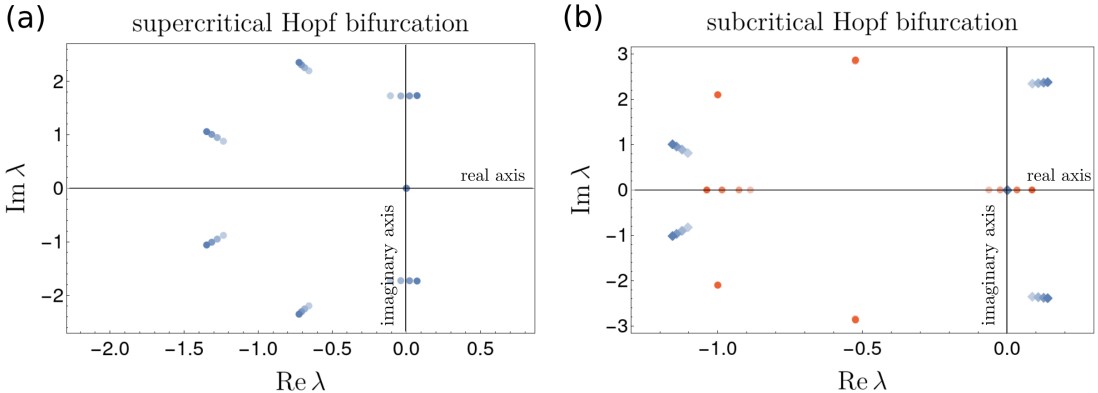

Figure 10: The behavior of eigenvalues of the Jacobian matrix at Hopf bifurcations, panels corresponding to the bottom row of Fig. 8. Eigenvalues corresponding to the equilibrium point with $S_z > 0$ are marked in blue, while red eigenvalues correspond to the equilibrium point with $S_z = 0$ and $S_x = 1/2$. Increasing intensity of points account for increasing $\eta$. (a) Supercritical Hopf bifurcation. A pair of complex conjugate eigenvalues cross the imaginary axis. The $S_z > 0$ stable equilibrium point become unstable, but a periodic orbit, i.e., a limit cycle, appears. (b) Subcritical Hopf bifurcation. As before, a pair of complex eigenvalues with Re $\lambda > 0$ is responsible for a limit cycle behaviour. Once the $S_z = 0$ equilibrium point loses its stability, i.e., when one of the real eigenvalues crosses the imaginary axis, the system jumps towards one of the stable limit cycle solutions.

to a single, $k = 0$ quasimomentum sector of the Hilbert space.

Here, now we consider the interacting problem with repulsive on-site intra-species ($V_1 > 0$) and inter-species ($V_2 > 0$) atomic interactions in a chain of $L = 51$ sites. We evaluate the contribution from nonzero quasimomenta by solving numerically the full Hamiltonian, but still in the mean-field regime, as in Ref. [26], due to the huge dimension of the Hilbert space. For simplicity, in the following we assume that $V_1 = V_2$, but we have checked that the results do not change qualitatively if we loose this assumption. We choose the initial state to be close

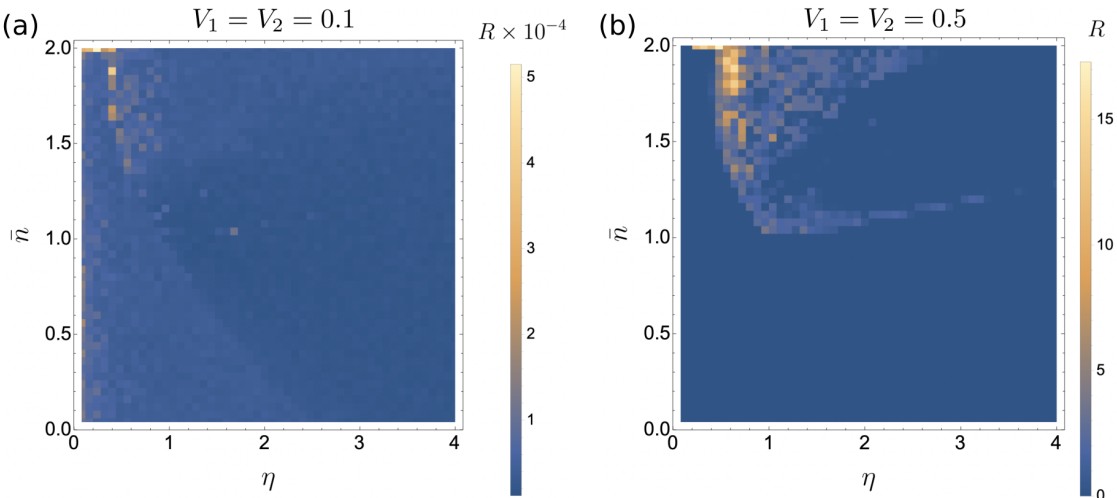

Figure 11: Density plots depicting the validity of the spin model, where we assume that small two-body contact interactions do not mix different quasi-momentum sectors of the Hilbert space significantly. We quantify the effect of the intra- ($V_1$) and inter- ($V_2$) atomic interactions on the single-mode assumption through the quantity $R$ defined in Eq. (33). (a) Negligible values of $R$ fully justify the single mode assumption for sufficiently small interactions, $V_1 = V_2 = 0.1$. (b) For stronger interactions, $V_1 = V_2 = 0.5$, on the other hand, the single-mode approximation is not valid in some parameter regimes as the contribution from higher quasimomentum states becomes important.

to a uniform, $k = 0$ plane wave, i.e.,

$$\psi_{\sigma,j}(t=0) = \mathcal{N}_\sigma \left(1 + r_j\right), \quad r_j \in [-\epsilon/2, \epsilon/2], \tag{32}$$

with $r_i$ being a random number from a uniform distribution centered around zero and the width $\epsilon = 2 \cdot 10^{-2}$. The normalization coefficient $\mathcal{N}_\sigma$ is calculated numerically in every disorder realization such that $\sum_j |\psi_{\sigma,j}|^2 = N_\sigma$. As before, we also assume that $\alpha(t=0)$ and $\beta(t=0)$ are random complex numbers lying within a complex circle of radius $\epsilon$. In order to check the robustness of our results we define a quantity

$$R = \max_{\sigma,\, t \in [T_1, T_2]} \left[ \frac{\sum_{k \neq 0} \left| \tilde{\psi}_{\sigma,k}(t) \right|^2}{\left| \tilde{\psi}_{\sigma,k=0}(t) \right|^2} \right], \tag{33}$$

which quantifies the maximal contribution of higher quasimomentum states due to the atom-atom interaction over the time evolution, where $\tilde{\psi}_{\sigma,k}(t)$ are the Fourier components of the wavefunction. As in the previous sections, the time interval $[T_1, T_2]$ can be arbitrarily chosen as long as $T_1, T_2, T_2 - T_1 \gg 1$.

According to the definition (33), the quantity $R$ is close to zero whenever the initial quasimomentum $k = 0$ is the dominant mode. In order to check the validity of the single quasimomentum assumption, we plot the quantity $R$ in Fig. 11 for different values of the interaction strength, $V_1 = V_2 = 0.1$ and $V_1 = V_2 = 0.5$. The other parameters are kept the same as before, i.e., $\kappa = 1$, $\Omega = 1$, and $\Delta = U_0 = -6$.

From Fig. 11(a), one can readily see that although $R$ is more sensitive in the region with unbroken $\mathbb{Z}_2$ symmetry, the single mode assumption is perfectly valid for sufficiently small interactions (note the overall scale of the plot). On the other hand, we find that for stronger

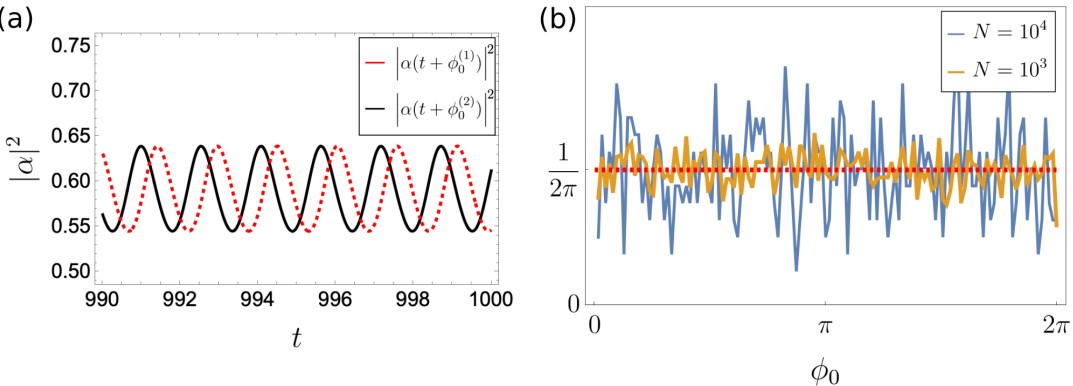

Figure 12: Time-dependent limit-cycle solutions spontaneously break the time-translation symmetry of the system. (a) A slight perturbation in initial conditions results in a solution that is shifted in time by a phase $\phi_0$. (b) The histogram of the phase $\phi_0$ distribution over many realizations converges towards a flat distribution, restoring the time translation symmetry.

atomic contact interactions the single mode assumption is not justified in some regions as illustrated in Fig. 11(b). Interestingly, it turns out that the higher quasimomentum modes are occupied most drastically in the dynamical limit cycle phases, which break the continuous time-translation symmetry of the system exhibiting time-crystalline-like behavior. We discuss this further in the next section.

## 6   Relation to time crystals

Time crystals are characterised by the spontaneous breaking of time transnational symmetry [45,46]. Although at first it seemed that coupling with the environment would always destroy the time crystal order, as in the case of many body localized (MBL) discrete time crystals (DTC) [47], it has been later realized that driven-dissipative atom-cavity systems not only can support the time crystal phase in perdiocially driven systems but also can stabilize it [48] as the dissipation prevents heating to the infinite temperature. Recently a first dissipative discrete time crystal has been realized in an optical cavity [49].

In this manuscript, the effective Hamiltonian (1) describing our system is time independent, i.e., it is invariant under the action of the continuous time translation operator. Nonetheless, as we have seen in the previous sections, the solutions in the two limit-cycle regimes break this continuous time translation symmetry by adopting periodic orbits $\alpha(t) = \alpha(t + T)$ with some period $T$. Indeed, it has been previously postulated that the regular limit cycle dynamics can be related to *continuous* time crystals [36, 37] (for the experiment see Ref. [39]), as long as the breaking of the symmetry is spontaneous. Spontaneous breaking of the continuous time symmetry means that if $\alpha(t)$ is a solution, so is $\alpha(t + \phi_0)$, where $\phi_0$ is an arbitrary phase shift and can be associated with a Goldstone mode [50]. In the process of spontaneous symmetry breaking the distribution of $\phi_0$ in many experimental (or, as in our case, numerical) independent realizations, slightly different in initial conditions, should be uniform. Indeed, in our case, the phase $\phi_0$ distribution in the limit-cycle phases tends to a flat distribution when the number of independent numerical simulations is increased. We confirm our findings in Fig. 12.

Last but not least, let us note that the occupation of higher quasimomentum modes (particularly in the limit cycles phases) due to the atomic interactions as discussed in Sec. 5 implies

the continuous space-translation symmetry of system is also broken. Combined with the broken time-translation symmetry in the limit cycle phases, this could lead to the emergence of space-time crystals. We plan to investigate this highly interesting scenario in the near future.

## 7 Summary and conclusions

In this work we have studied nonequilibrium phases of a quasi-1D two-component BEC in a transversely pumped two-mode linear cavity with cavity-generated dynamical gauge fields. Our system can be described by a two-leg Bose-Hubbard model with leg-dependent, cavity-assisted, dynamical complex tunneling amplitudes. Our effective lattice model constitutes a minimal, dynamical flux-lattice model with only two lattice sites in the transverse (rung) direction. A comprehensive understanding of a minimal model is important in the investigation of more complicated ones in order to set up a framework for the future research directions.

Using the effective spin mapping of the bosonic operators, we have described the full dynamical phase diagram of the model and found a plethora of non-stationary phases, such as two distinct limit-cycle phases and a chaotic phase. Specifically, we have found a Feigenbaum-like period doubling leading to chaos and strange attractors. We have also investigated the equilibrium points of equations of motion as well as their stability, which provides a complementary understanding of the dynamical phases of the system. The analysis of the eigenvalues of the Jacobian matrix at equilibrium points has allowed us to recognize pitchfork and Hopf bifurcations of the dynamical solutions. Finally, we have studied the robustness and validity regime of our findings and discussed the relation of limit-cycle solutions to time crystals.

Last but not least, we would like to stress that although our considerations in this paper are focused entirely on weakly interacting ultracold atoms in optical lattices where a collective spin mapping is justified, our methods and results could be of applicable to other experimental systems (e.g. Rydberg atoms [51] and trapped ions [52, 53]) where, on the contrary, strong global-range interactions [54, 55] makes the single spin approximation viable, as in Lipkin-Meshkov-Glick (LMG) [56] and similar models.

## Acknowledgements

We thank Elvia Coella for fruitful discussions. A.K. and H. R. acknowledge support from the FET Network Cryst3 funded by the European Union (EU) via Horizon 2020. F. M. acknowledges financial supports from the Stand-alone Project P 35891-N of the Austrian Science Fund (FWF), and the ESQ-Discovery Grant of the Austrian Academy of Sciences (ÖAW).

## A Appendix

In this work the parameters of the Hamiltonian, Eq. (6), has been chosen arbitrarily as $\kappa = 1$, $\Omega = 1$ and $\Delta = U_0 = -6$ with the initial state $\vec{S} = (\frac{1}{2}, 0, 0)$, so that the results of this paper could be directly compared with the results from our previous Letter, Ref. [26]. In this Appendix we plot dynamical phase diagrams for different parameters [Fig. 13] and initial conditions [Fig. 14], showing the same dynamical phases with quantitative differences only.

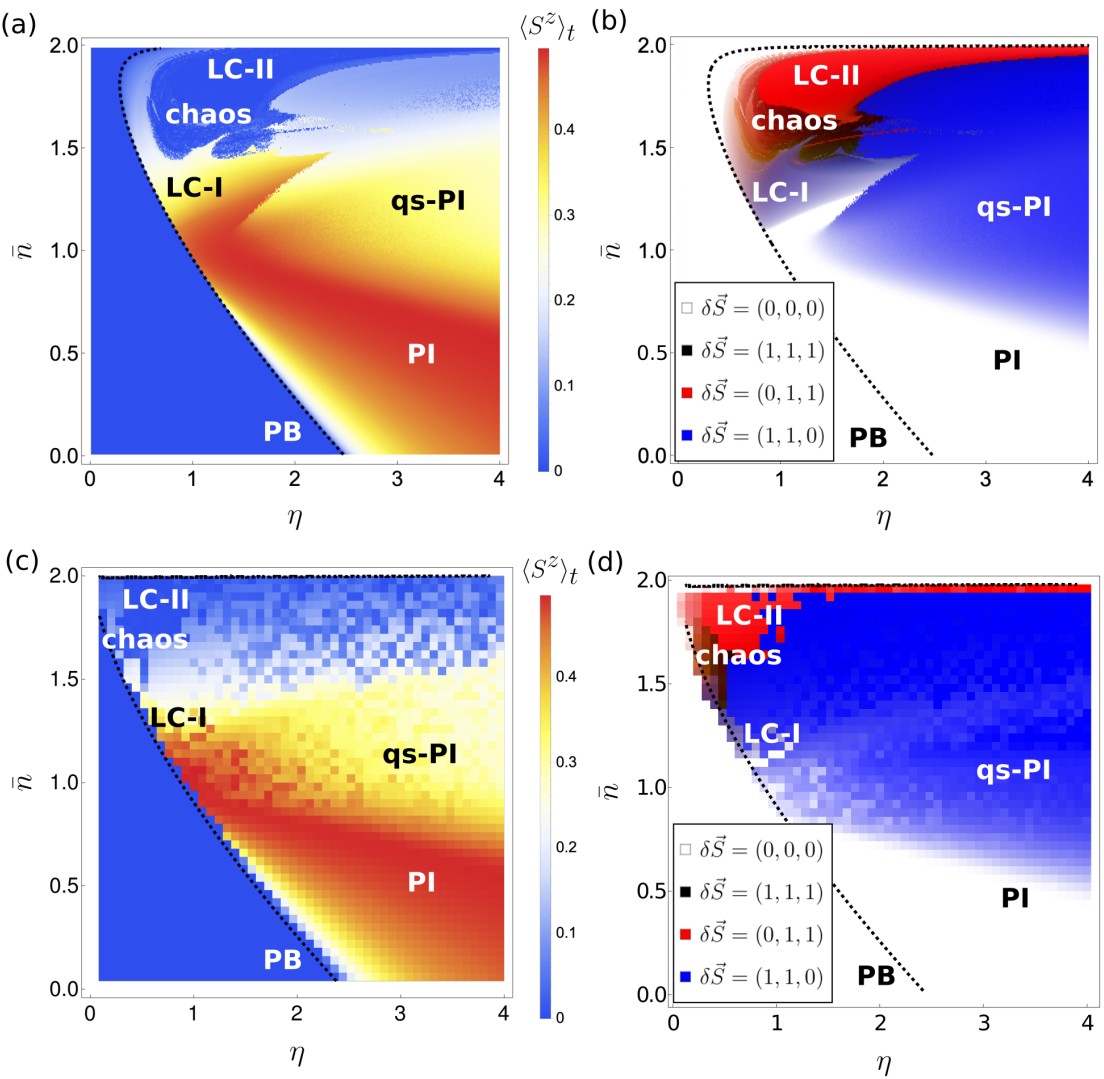

Figure 13: Dynamical phase diagram showing the long-time behavior of the model for $\kappa = 1$, $\Omega = 0.1$, $\Delta = U_0 = -6$ [(a) and (b)], and $\kappa = 0.25$, $\Omega = 1$, $\Delta = U_0 = -6$ [(c) and (d)]. The panels of the left column [(a) and (c)] show the $z$-component of the collective spin $\langle S^z \rangle_t$ while the panels of the right column [(b) and (d)] display the vector of maximum spin fluctuations $\delta \vec{S}$ using the CMY color model [cf. Fig. 2 and the discussion in Sec. 3 of the main article]. The initial state is $\vec{S} = (\frac{1}{2}, 0, 0)$.

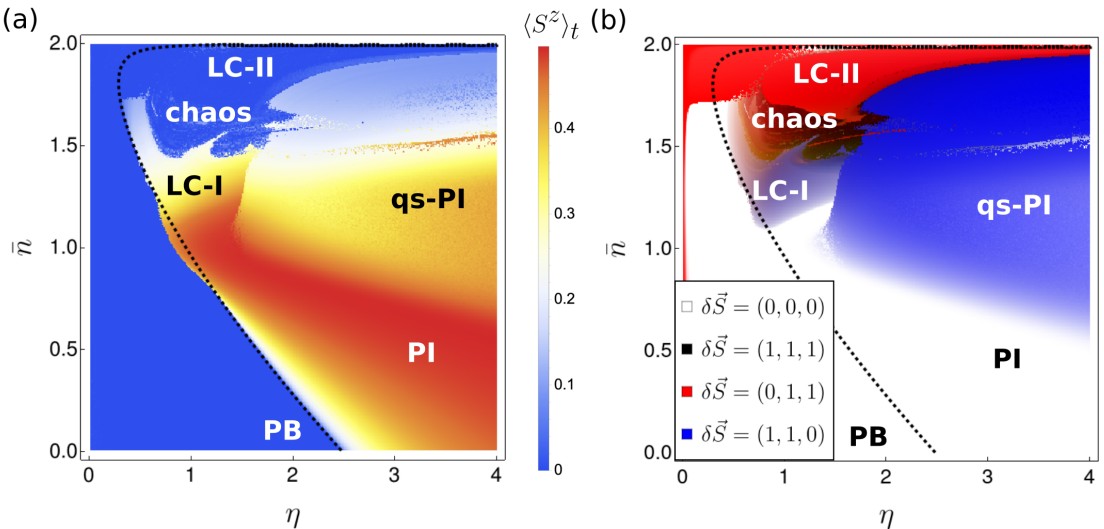

Figure 14: Dynamical phase diagram showing the long-time behavior of the model for the initial state $\vec{S} = (0, 0, \frac{1}{2})$. The other parameters of the model are chosen as in figures in the main text, i.e., $\kappa = 1$, $\Omega = 0.1$, $\Delta = U_0 = -6$. (a): The $z$-component of the collective spin $\langle S^z \rangle_t$. (b): The vector of maximum spin fluctuations $\delta\vec{S}$ using the CMY color model [cf. Fig. 2 and the discussion in Sec. 3 of the main article].

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
