# Peer review of "Nonequilibrium phases of ultracold bosons with cavity-induced dynamic gauge fields"

_SciPost Physics_

## Round 1 · Referee Report · Anonymous · 2022-10-31

Strengths
-) Highly relevant topic.
-) Strong connection with cavity QED experiments.
-) Exhaustive and clear presentation of the results.
Weaknesses
-) Lack of substantial novelty.
-) Literature context missing.
Report
The authors study the dynamical phase diagram and the quasi-stationary phases of a one dimensional two leg ladder Hubbard model. This study follows the proposal of the experimental realization of the model by the same authors in Ref. [26], where the stationary phases of the Hamiltonian under consideration are also studied.
The current investigations are performed by mapping the bosonic operators into a single global spin, a procedure which disregards both inter and intra species two body interactions. Thus, the authors are left with the task of studying a single N-component spin variable. Since they are interested in the thermodynamic limit, they disregard the quantum nature of the spin operators, resulting in a system of coupled dynamical equations, see Eqs. (10) and (11) that can be studied with the traditional tools of dynamical systems.
The nature of the phase diagram is reported in Fig. 2 and, apart from the stationary states already described in Ref.[26] , features quasi-stationary states and two limit cycles phases. I don't find these findings particularly relevant or interesting \emph{per se}. In the end, similar phenomena emerge ubiquitously in classical dynamical system like Eqs. (10) and (11) and, so, I don't find these claims particularly novel. The novelty should in principle came from the relation with the cavity QED platforms into consideration, but this has already been discussed in Ref. [26].
Having said that, the paper is well written. It makes a thorough analysis and clarifies many details of the discussion made in Ref. [26], which will have some value for researches approaching the field or trying to repeat a similar calculation.
However, in order to accomplish this "pedagogical" task, the paper shall also make appropriate reference to the literature not only in the cavity QED field, but also in various other cold atom platforms, where similar calculations may be useful. Therefore, I suggest the authors to include a discussion on how their results may be applicable to other experimental systems, where the global interaction range makes the single spin approximation viable. I am referring to Rydberg atom, trapped ions and long-range systems in general. The authors may refer to three recent reviews on these topics:
[1] C. Monroe, et al. Rev. Mod. Phys. 93, 025001 (2021).
[2] N. Defenu, et al. arXiv:2109.01063 (2021).
[3] L. Chomaz, et al. arXiv:2201.02672 (2021).
Moreover, I found that several traditional topics in dynamical systems are quoted without making reference to proper literature. In particular, the concepts of "subcritical pitchfork", "Hopf bifurcation" and, even, "Landau theory" appear in the manuscript without any proper introduction nor quotation to references. These are traditional topics in classical dynamical systems, but not all the members of the quantum physics community shall be familiar with these concepts.
The last section on time-crystals is also a bit superficial. In the beginning it is stated that coupling to an environment was expected to be detrimental to the formation of time crystalline phases, but this was proven not to apply to cavity QED systems. I expect this to be due to the long-range coherent interactions generated by the cavity. Indeed, it has been shown that long-range interactions play a crucial role in the stabilization of time crystal phases. Maybe the authors can comment on the importance of long-range interactions in the stabilization of time crystal in their model and, also, consider recent studies where the relation between time crystal and dynamical systems has been considered:
[4] Pizzi et al. Nat. Comm. 12, 2341 (2021) .
[5] M. Collura, et al. Phys. Rev. X 12, 031037 (2022).
[6] G. Giachetti, et al. arXiv:2203.16562 (2022).
All the calculations and the claims made by the papers appear to be very correct. I have however one doubt regarding Sec. 5 "Validity of the spin model". There, the authors discuss the validity of their picture with respect to the inclusion of a two-body interaction potential between atoms of the same species. The plot in Fig. 11 are rather comforting since they show that for small enough interactions the single mode approximation is actually stable to the inclusion of atom-atom interactions. Yet, I do not understand why the authors did not include an interaction term between the two atomic species. Is there any reason to believe this term shall be less dangerous than the intra-species one? Can the authors comment on this point?
Requested changes
1) Include a discussion on different quantum platforms which may feature the dynamical phase diagram under study.
2) Define more extensively concepts of dynamical systems used in the papers, such as "subcritical pitchfork", "Hopf bifurcation" and, even, "Landau theory", making reference to proper literature.
3) Extend the discussion of time crystal phases in relation to the presence of long-ranfe interactions in the model.
4) Discuss the impact of inter-species interactions in addition to intra-species ones.
Author: Arkadiusz Kosior on 2022-12-14 [id 3133]
(in reply to Report 1 on 2022-10-31)
We thank the Referee for the overall positive evaluation of our manuscript and the feedback. In the following we answer all the raised questions one by one.
The referee writes:
The authors study the dynamical phase diagram and the quasi-stationary phases of a one dimensional two leg ladder Hubbard model. This study follows the proposal of the experimental realization of the model by the same authors in Ref. [26], where the stationary phases of the Hamiltonian under consideration are also studied.
The current investigations are performed by mapping the bosonic operators into a single global spin, a procedure which disregards both inter and intra species two body interactions. Thus, the authors are left with the task of studying a single N-component spin variable. Since they are interested in the thermodynamic limit, they disregard the quantum nature of the spin operators, resulting in a system of coupled dynamical equations, see Eqs. (10) and (11) that can be studied with the traditional tools of dynamical systems.
The nature of the phase diagram is reported in Fig. 2 and, apart from the stationary states already described in Ref.[26] , features quasi-stationary states and two limit cycles phases. I don't find these findings particularly relevant or interesting per se. In the end, similar phenomena emerge ubiquitously in classical dynamical system like Eqs. (10) and (11) and, so, I don't find these claims particularly novel. The novelty should in principle came from the relation with the cavity QED platforms into consideration, but this has already been discussed in Ref. [26].
Having said that, the paper is well written. It makes a thorough analysis and clarifies many details of the discussion made in Ref. [26], which will have some value for researches approaching the field or trying to repeat a similar calculation.
However, in order to accomplish this "pedagogical" task, the paper shall also make appropriate reference to the literature not only in the cavity QED field, but also in various other cold atom platforms, where similar calculations may be useful. Therefore, I suggest the authors to include a discussion on how their results may be applicable to other experimental systems, where the global interaction range makes the single spin approximation viable. I am referring to Rydberg atom, trapped ions and long-range systems in general. The authors may refer to three recent reviews on these topics:
[1] C. Monroe, et al. Rev. Mod. Phys. 93, 025001 (2021). [2] N. Defenu, et al. arXiv:2109.01063 (2021). [3] L. Chomaz, et al. arXiv:2201.02672 (2021).
Our response:
As suggested by the Referee, in the new version of the manuscript we mentioned different cold-atom platforms with global range interactions and added the references to the corresponding works to our reference list.
The referee writes:
Moreover, I found that several traditional topics in dynamical systems are quoted without making reference to proper literature. In particular, the concepts of "subcritical pitchfork", "Hopf bifurcation" and, even, "Landau theory" appear in the manuscript without any proper introduction nor quotation to references. These are traditional topics in classical dynamical systems, but not all the members of the quantum physics community shall be familiar with these concepts.
Our response:
We agree with the Referee that some concepts might not have been introduced precisely enough in our manuscript. The new version includes both references to the classical literature and, for the sake of being self-contained, explain the aforementioned concepts for classical dynamical systems.
The referee writes:
The last section on time-crystals is also a bit superficial. In the beginning it is stated that coupling to an environment was expected to be detrimental to the formation of time crystalline phases, but this was proven not to apply to cavity QED systems. I expect this to be due to the long-range coherent interactions generated by the cavity. Indeed, it has been shown that long-range interactions play a crucial role in the stabilization of time crystal phases. Maybe the authors can comment on the importance of long-range interactions in the stabilization of time crystal in their model and, also, consider recent studies where the relation between time crystal and dynamical systems has been considered.
[4] Pizzi et al. Nat. Comm. 12, 2341 (2021) . [5] M. Collura, et al. Phys. Rev. X 12, 031037 (2022). [6] G. Giachetti, et al. arXiv:2203.16562 (2022).
Our response:
First of all, originally we decided to include a section on time crystals in our manuscript because of the highly relevant recent experimental works on dissipative continuous time crystals as e.g. in Kongkhambut et al, Science 377, 670-673 (2022). Secondly, let us point out that the examples given by the Referee consider discrete time crystals in closed systems, where long-range interactions are important to stabilise disorder-free models. In our case the gauge potential and whole dynamics arise due to the cavity decay, and not due to an external driving. In open systems as in our case, it is the dissipation that prevents heating to the infinite temperature and, hence, stabilises the time crystals behaviour [see the latest experimental works: Kessler et al, Rev. Lett. 127, 043602 (2021) and Taheri et al, Nature Communications 13, 848 (2022)]. We expand and clarify this issue in the new version of our manuscript.
The referee writes:
All the calculations and the claims made by the papers appear to be very correct. I have however one doubt regarding Sec. 5 "Validity of the spin model". There, the authors discuss the validity of their picture with respect to the inclusion of a two-body interaction potential between atoms of the same species. The plot in Fig. 11 are rather comforting since they show that for small enough interactions the single mode approximation is actually stable to the inclusion of atom-atom interactions. Yet, I do not understand why the authors did not include an interaction term between the two atomic species. Is there any reason to believe this term shall be less dangerous than the intra-species one? Can the authors comment on this point?
Our response:
We agree with the Referee that this issue might not have been well explained in the previous version, where we decided to include only interspecies interactions for simplicity reasons as interspecies interactions did not change the results qualitatively (see the attached figure). In the new version of the manuscript we include both inter- and intra- species interactions. From this simulation we can see that the main conclusion that the single quasi-momentum approximation is valid for small enough contact atomic interactions remains still intact.
Author: Arkadiusz Kosior on 2022-12-14 [id 3134]
(in reply to Report 2 on 2022-11-05)We thank the Referee for the overall positive evaluation of our manuscript and her/his comments.
The referee writes:
Our response:
We would like to thank the Referee again for the appreciation of our work.
The referee writes:
Our response:
The Referee is correct. Usually, before ultracold atoms are loaded into an optical lattice, a BEC is prepared in a spatially uniform ground state in a zero momentum state. We expand on this issue in the new version of the manuscript.
The referee writes:
Our response:
We evaluate the contribution from nonzero quasimomenta by solving numerically the full Hamiltonian, but still in the mean-field regime, as in Ref. [26], due to the huge dimension of the Hilbert space. We expand on this in the new version of the manuscript.
The referee writes:
Our response:
The Referee is right that in principle exact many-body quantum calculations could have non-neglible effects, especially for a strongly interacting system. Indeed, we plan to include many-body techniques in a future work in strongly interacting regime. However, as long as we are dealing with large photon numbers, an anihilation of a single photon does not change substantially the mean number of photons, and therefore, a coherent state approximation for photons is well justified. It is also true for non- (or weakly) interacting atoms in a thermodynamic limit, where corrections to the mean-field are suppressed as $1/V$ [see for example F. Mivehvar et at, Advances in Physics, 70, 1 (2021)]. We elaborate on this in the new version of the manuscript.

---

## Round 1 · Referee Report · Anonymous · 2022-11-5

Strengths
The authors study interesting mean-field properties of a 1D system of bosons, coupled to two types of photons, with different detunings. Such coupling leads to an effective 2D description of the system. They show that the system can be mapped into a collective spin problem. They derive the Heisenberg equations of motion for the spin and photon operators and treat them classically to investigate the chaotic/non-chaotic nature of the derived equations of motion. I found their findings interesting and believe it would be interesting for the readers of the journal as well.
Weaknesses
However, there are a few points that the authors could clarify before the publication of their work:
First, can the authors elaborate more on why $k=0$ is experimentally the most relevant mode? Is it assumed the system starts with a BEC?
Second, in section 5, how do the authors evaluate the contribution from nonzero momenta, $k\neq 0$? It would be clarifying if they can provide a more detailed argument of how they calculate/simulate the correction due to these modes. I assume it is not a full quantum simulation for L=51 sites and a/b dynamical bosons.
Third, from the very beginning, the authors have argued that only the mean-field value of the spin and photon field operators can be retained in the equations and they can be treated as classical entities. Is it possible for the authors to justify this claim? More specifically, there is the possibility of doing expansion beyond the mean-field analysis to investigate how big the corrections are; and in what regime of parameters the aforementioned mean-field analysis is valid.
Report
I found their findings interesting and believe it would be interesting for the readers of the journal as well.

---

## Editorial Decision

unknown